# Pan-Cancer Analysis and Experimental Validation Identify ACOT7 as a Novel Oncogene and Potential Therapeutic Target in Lung Adenocarcinoma

**DOI:** 10.3390/cancers14184522

**Published:** 2022-09-18

**Authors:** Chao Zheng, Guochao Zhang, Kai Xie, Yifei Diao, Chao Luo, Yanqing Wang, Yi Shen, Qi Xue

**Affiliations:** 1Department of Thoracic Surgery, National Cancer Center/National Clinical Research Center for Cancer/Cancer Hospital, Chinese Academy of Medical Sciences and Peking Union Medical College, Beijing 100021, China; 2Department of Cardiothoracic Surgery, Jinling Hospital, Medical School of Southeast University, Nanjing 210009, China; 3Department of Cardiothoracic Surgery, Jinling Hospital, Nanjing Medical University, Nanjing 210009, China; 4Department of Cardiothoracic Surgery, Jinling Hospital, Medical School of Nanjing University, Nanjing 210009, China; 5Department of Cardiothoracic Surgery, Jinling Hospital, Southern Medical University, Guangzhou 510515, China; 6Department of Cardiology, Jinling Hospital, Nanjing University, Nanjing 210009, China

**Keywords:** ACOT7, pan-cancer, LUAD, immune microenvironment, prognosis

## Abstract

**Simple Summary:**

Acyl-CoA thioesterase 7 (ACOT7) is important in regulating cell cycle, cell proliferation, and glucose metabolism. Research on the functions of ACOT7 are seldom, and comprehensive pan-cancer analysis is lacking. We aimed to perform a pan-can analysis and validated the prognostic value of ACOT7 in lung adenocarcinoma. ACOT7 was tightly associated with the tumor microenvironment. The downregulation of ACOT7 expression suppressed cell proliferation and the migration of the PC9 cell line. ACOT7 is a novel oncogene and therapeutic target for lung adenocarcinoma.

**Abstract:**

Background: Acyl-CoA thioesterase 7 (ACOT7) is of great significance in regulating cell cycle, cell proliferation, and glucose metabolism. The function of ACOT7 in pan-cancer and its capacity as a prognostic indicator in lung adenocarcinoma (LUAD) remains unknown. We intended to perform a comprehensive pan-cancer analysis of ACOT7 and to validate its value in LUAD. Methods: The expression levels, prognostic significance, molecular function, signaling pathways, and immune infiltration pattern of ACOT7 in 33 cancers were explored via systematic bioinformatics analysis. Multivariate Cox regression was applied to construct nomograms to predict patients’ prognoses. Moreover, we conducted in vitro experiments including CCK8, scratch, Transwell, and Matrigel assays to further explore the function of ACOT7 in LUAD. Results: Patients with high ACOT7 expression have notably poorer long-term survival in many cancer types, including LUAD. Further enrichment analyses reveal that ACOT7 is involved in immune cells’ infiltration and is substantially related to the cancer–immune microenvironment. ACOT7 could influence drug sensitivities, including afatinib, gefitinib, ibrutinib, lapatinib, osimertinib, sapitinib, taselisib, and PLX-4720 (all *p* < 0.01). A nomogram demonstrated a fair predictive value of ACOT7 in LUAD (C-index: 0.613, 95% CI: 0.568–0.658). The proliferation and migration of PC9 cells were significantly repressed when ACOT7 expression was downregulated. Conclusion: As an oncogene, ACOT7 is critical in the tumor microenvironment of pan-cancer and might be a novel therapeutic target for LUAD.

## 1. Introduction

Acyl-CoA thioesterase 7 (ACOT7) is one of the most widely investigated isoforms of the ACOT family [1], also termed ACT, ACH1, BACH, LACH, LACH1, and CTE-II. The enzyme is localized in the cytoplasm and is expressed in various tissues, particularly in brain tissue and testes [2,3]. Moreover, ACOT7 is involved in the hydrolysis process of arachidonoyl-CoA and supplies sufficient arachidonic acid to synthesize prostaglandins [4]. As an essential precursor molecule for pro-inflammatory eicosanoids, arachidonic acid is essential in regulating cell cycle, cell proliferation, and glucose metabolism [1,5]. Forwood et al. reported that ACOT7 expression in macrophages is upregulated by lipostagelandins [1]. ACOT7 has also been reported to be involved in cell cycle control as well as a target for therapies in breast and lung cancer [6]. Although ACOT7 has been reported in some types of cancer, there are still many gaps in the research across the cancer spectrum.

Cancer is a critical health burden worldwide, and lung cancer has the highest mortality rate [7]. Unfortunately, there are still no effective therapies to cure cancer. In recent years, the field of immunotherapy has emerged to be a powerful and promising therapy for treating various cancer [8]. Many studies have been conducted to explore new immunotherapeutic targets by integrating public data for pan-cancer expression and survival analysis [9].

The tumor microenvironment (TME) provides an indispensable function in cancer progression [10]. The TME contains many cellular components and surrounding non-cellular components closely related to tumor progression, and it has become a therapeutic target [11]. Platelet- and tumor-associated macrophages can directly facilitate cancer growth, migration, and metastasis [12,13]. The neutrophil-to-lymphocyte ratio in the TME is related to lung cancer prognosis [14]. Currently, immunotherapy is a rapidly growing area of cancer research. However, not all patients are sensitive to programmed death-1 (PD-1) and programmed death ligand-1 (PD-L1) blockade therapies, and there are even patients who are resistant to inhibitors [15,16], although this appears to be a promising strategy for cancer immunotherapy [17,18,19]. Finding a novel therapeutic target to compensate for a shortage of PD-1/PD-L1 antibodies might be a promising approach to reduce advanced cancer recurrence.

Some research has reported ACOT7’s role in specific tumors. Pan-cancer analysis on the role of ACOT7 is lacking. Therefore, we aimed to explore the expression level, prognostic significance, molecular function, signaling pathways, and immune infiltration pattern of ACOT7 in 33 types of cancer via systematic bioinformatics analysis and perform functional experimental validation in LUAD.

## 2. Materials and Methods

### 2.1. Differential Expression of ACOT7 in Pan-Cancer

The Cancer Genome Atlas (TCGA, https://tcga-data.nci.nih.gov/tcga/, accessed on 5 March 2021) is a publicly accessible, web-based database that offers gene expression, copy number, mutation, and clinical information [20,21]. The Genotype-Tissue Expression Project (GTEx, accessed on 5 March 2021) [22] has nearly 1000 individuals with 53 normal tissue loci for RNA sequencing, and RNA-seq data can be obtained from the University of California, Santa Cruz Xena (UCSC Xena, accessed on 5 March 2021) [23]. The Cancer Cell Line Encyclopedia (CCLE, accessed on 5 March 2021) [24] was applied to obtain the expression level data for cell lines. Data from the TCGA, GTEx, and CCLE databases were combined using the “limma” (version 4.1) package of R software (version 4.0.2, https://www.R-project.org, 1 July 2020, The Comprehensive R Archive Network, open source). ACOT7 expression was assessed in 33 cancers, 31 normal tissues, and 31 tumor cell lines using the downloaded data. Expression data were Log_2_ transformed. The expression levels of ACOT7 were compared between cancer samples and matched standard samples in 33 cancers using two sets of *t*-tests, with *p* < 0.05 indicating significant differential expression between tumor and normal tissues. The Human Protein Atlas (HPA, https://www.proteinatlas.org/, accessed on 5 March 2021) [25,26] was applied for the analysis of the protein expression of ACOT7 in human tissue and cells via immunohistochemical images.

### 2.2. ACOT7 and Prognosis Analyses in Pan-Cancer

The relationship between ACOT7 expression and long-term survival in cancers was explored using TCGA. The expression of ACOT7 was divided into two groups according to the optimal cutoff value via X-tile 3.6.1 software. Kaplan–Meier curves based on univariate Cox analyses were applied when assessing the prognostic value of ACOT7. The log-rank test was applied to compare the differences between groups. Forest plots were used to visualize the relationship between ACOT7 and overall survival (OS), disease-free interval (DFI), disease-specific survival (DSS), and progression-free interval (PFI) in different cancers based on “survival” (version 3.3) and “forestplot” (version 3.6) packages of R software.

### 2.3. Construction of Nomogram in LUAD

To better predict the prognosis of patients with LUAD, we further obtained clinical features of patients from the TCGA database [27]. Multivariate Cox regression analysis was conducted using the enter method. Important variables, including gender, age, smoking status, and race, which were reported as risk factors for the worse survival of patients with lung cancer, were included in the regression model [28,29,30,31,32]. The potential multicollinearity of the model was estimated via the variance inflation factor (VIF), with VIF ≥ 5 indicating significant multicollinearity. Then, the predictive model was visualized using a nomogram. The C-index and calibration plots were applied to evaluate the predictive accuracy of the nomogram. The C-index ranges from 0 to 1.0, with 1.0 indicating perfect predictive accuracy, 0.8–1.0 indicating excellent accuracy, and 0.6–0.8 indicating fair accuracy. Patients were placed into low- or high-risk groups based on the scores calculated from the nomogram. The optimal cutoff values of ACOT7 expression and scores were identified via X-tile 3.6.1 [33].

### 2.4. Genetic Alteration Analysis of ACOT7

The cBioPortal database (www.cbioportal.org, accessed on 5 March 2021) [34,35] was applied to obtain copy number alteration (CNA) and mutation data of ACOT7 and to analyze the genomic alterations pattern of ACOT7 in tumors. The Person’s correlation between CNA, DNA methylation levels, and ACOT7 expression was further analyzed.

### 2.5. Correlation between ACOT7 Expression and TME

Gene sets with significant impact were selected using gene set enrichment analysis (GSEA) at a false discovery rate less than 0.05 and ranked using normalized enrichment scores. Gene set variation analysis (GSVA) using the MSigDB database (v 7.1, accessed on 12 March 2021) [36] was also performed to further explore the immune-cell-related pathways which ACOT7 correlated with. Then, we applied CIBERSORT [37], a bioinformatic algorithm, to calculate the composition of immune cells and to assess immune infiltration patterns. TISIDB, an integrated repository portal for tumor-immune system interactions, was utilized to analyze the relationship between ACOT7 expression and immune-related genes using heatmaps [38]. Relationships between ACOT7 expression and tumor mutational burden (TMB), microsatellite instability (MSI), and immune-checkpoint-associated genes were also investigated using Pearson’s correlation analyses.

### 2.6. Drug Screening of ACOT7

Genomics of Drug Sensitivity in Cancer (GDSC, version 2, https://www.cancerrxgene.org/, accessed on 20 March 2021) [39] was used to explore the relationship between ACOT7 expression and the half maximal inhibitory concentrations (IC50s) of 198 compounds using Spearman’s correlation analyses [40]. In addition, the difference in IC50 concentrations between the high and low ACOT7 expression groups was determined using *t*-test.

### 2.7. Cell Culture and siRNAs Transfection

Human bronchial epithelial (HBE) cells and lung cancer cell lines, including PC-9 (RRID: CVCL_B260), A-549 (RRID: CVCL_0023), and NCI-H1975 (RRID: CVCL_1511), were purchased from Shanghai Institutes of Biological Sciences, China. Cells were all cultured in DMEM (KeyGene, Nanjing, China) and maintained in a humidified incubator at 37 °C with 5% CO_2_. The PC9 cell line was transfected by siRNAs (RiboBio, Guangzhou, China) using Lipofectamine RNAiMAX (Invitrogen, Carlsbad, CA, USA). The sequences of siRNAs targeting ACOT7 were 5′-AGACCGAGGACGAGAAGAADTDT-3′ (si1-ACOT7) and 5′-GUGCAGGUCAACGUGAUGUDTDT-3′ (si2-ACOT7). The negative control (NC) siRNA sequence was: 5′-GCACCCAGTCCGCCCTGAGCAAATTCAAGAGATTTGCTCAGGGCGGACTGGGTGCTTTTT-3′.

### 2.8. RNA Extraction

Trizol reagents (Invitrogen) were used to extract total RNA from PC9 cells. ACOT7 expression was measured via qRT-PCR using the SYBR Select Master Mix (Applied Biosystems, Cat: 4472908). Reverse transcription was performed to generate cDNA using a Reverse Transcription Kit (Takara, Cat: RR036A, KeyGEN). qPCR conditions were as follows: 10 min at 95 °C followed by 35–40 cycles at 95 °C for 15 s and 60 °C for 34 s, followed by a plate read after each cycle. Relative RNA levels were calculated using the comparative 2-ΔΔCT method with GAPDH as an endogenous control [41]. The primer sequences of GAPDH were as follows: forward: 5′-ATGGGGAAGGTGAAGGTCG-3′; reverse: 5′-CTCCACGACGTACTCAGCG-3′. The primer sequences of ACOT7 were: forward: 5′-TCTCCCATGTGCATCGGTG-3′; reverse: 5′-TTTTCGGACATCACGTTGACC-3′. RNA concentrations were determined using a microplate reader (Oy spectrophotometer 1510, Thermo Fisher Scientific). qPCR was performed on an Applied Biosystems 7900 qRT-PCR machine (Applied Biosystems, ThermoFisher Scientific, Foster City, CA, USA).

### 2.9. Cell Proliferation and Migration/Invasion Assays

PC9 cells were transfected with siRNAs overnight using 96-well plates at the control density (2000 cells per well) and incubated for 2 h with CCK-8 (10 μL/well) in each well. The reaction products were measured at 450 nm. A scratch wound-healing migration assay was performed. When the cells reached 90–100% confluence in the 6-well plate, the tip of a sterile plastic pipette was scraped in each culture well, washed twice with PBS to remove cell debris, and then, the cells were placed in serum-free medium for 24 h. Images were captured via fluorescence microscopy using an inverted microscope (Nikon Eclipse 2000, Nikon, Tokyo, Japan). We inoculated 5 × 10^4^ cells onto Transwell kits (8 mm PET, 24-well Millicell) or matrix-coated inserts (BD Biosciences, Bedford, MA, USA) and incubated them for 24 h or 48 h. Cells were counted under an inverted microscope after being fixed in methanol and stained with 0.1% crystal violet.

### 2.10. Statistical Analysis

Gene expression data were normalized by log_2_ transformation. The Shapiro–Wilk test was used to test the normality of data. When comparing gene expression between normal and cancer tissues, the t-test or Wilcoxon test was used as appropriate. Multivariate Cox analysis was used to construct the nomograms. The survival difference between the groups was visualized via Kaplan–Meier curves. Pearson’s or Spearman’s test was used to assess the strength of the correlations between variables, as appropriate. The means with standard deviation (SD) were used to compare the relative expression level of ACOT7 in groups when analyzing the results of the in vitro experiments. R software (version 4.0.2, https://www.R-project.org, 1 July 2020, The Comprehensive R Archive Network, open source) and GraphPad Prism (version 8.0.2, GraphPad Software, Inc., San Diego, CA, USA) were used for statistical analysis and visualization. A two-sided *p* < 0.05 was a statistical threshold.

## 3. Results

### 3.1. Expression Levels of ACOT7 in Pan-Cancer

ACOT7 expression in 33 cancers was systematically analyzed (Figure 1A). Appendix A summarizes information regarding the characteristics of the 33 cancers. ACOT7 was highly expressed in 25 cancers. As presented in Figure 1B, ACOT7 was expressed in all tumors, with the highest expression level being in SKCM and the lowest expression level being in KICH. ACOT7 was relatively highly expressed in key physiological tissues such as pituitary, brain, bone marrow, and testis tissue (Figure 1C). In addition, CCLE analysis suggested that ACOT7 was expressed in all tumor cell lines (Figure 1D). Moreover, the HPA database verified that ACOT7 protein expression levels and mRNA expression were consistent (Appendix A). Collectively, the above results demonstrated that ACOT7 is aberrantly expressed in various cancers.

### 3.2. ACOT7 as a Prognostic Indicator in Cancers

ACOT7 expression was related to the tumor node metastasis (TNM) stage in six cancers: HNS, KIRC, KIRP, LIHC, LUAD, and THCA (Appendix A). Moreover, compared with paired adjacent normal tissue, ACOT7 was highly expressed in BLCA, BRCA, HNSC, KIRC, LIHC, LUSC, STAD, and THCA and significantly decreased in KICH and KIRP (Appendix A).

The correlation between long-term survival and ACOT7 mRNA expression in pan-cancer was explored. ACOT7 expression correlates with patients’ OS in 10 cancers, including ACC (hazard ratio (HR) = 1.624, 95% confidence interval (CI): 1.190–2.216, *p* = 0.002), BLCA (HR = 1.189, 95% CI: 1.012–1.396, *p* = 0.035), GBM (HR = 1.294, 95% CI: 1.033–1.620, *p* = 0.025), KIRP (HR = 1.297, 95% CI: 1.012–1.661, *p* = 0.040), LAML (HR = 1.481, 95% CI: 1.156–1.896, *p* = 0.002), LIHC (HR = 1.530, 95% CI: 1.239–1.889, *p* = 0.001), LUAD (HR = 1.296, 95% CI: 1.077–1.560, *p* = 0.006), LUSC (HR = 1.199, 95% CI: 1.003–1.433, *p* = 0.046), MESO (HR = 2.007, 95% CI: 1.452–2.772, *p* < 0.001), and OV (HR = 0.882, 95% CI: 0.792–0.982, *p* = 0.022) (Figure 2A and Appendix A). In addition, ACOT7 expression was closely related to DSS (Figure 2B), DFI (Figure 2C), and PFI (Figure 2D) in several cancers, especially lung cancer. Collectively, ACOT7 is a prognostic biomarker in various cancers, especially in ACC, KIRP, LIHC, LUSC, and MESO.

A total of 515 patients with LUAD from the TCGA database were extracted to verify the value of ACOT7. Table 1 shows the results of the Cox regression models. A nomogram including ACOT7 expression, race, gender, T stage, N stage, and prior malignancy was developed to predict the OS (Figure 3A). Age was not included in the final model due to the multicollinearity. The calibration plots demonstrated a fair predictive accuracy for predicting OS with a C-index of 0.613 (95% CI: 0.568–0.658) (Figure 3B). The high-risk group (score > 225) demonstrated a dramatically poorer OS (HR = 2.10, 95% confidence interval (CI): 1.54–2.87, *p* < 0.001) (Figure 3C).

### 3.3. Genetic Alteration Analyses of ACOT7

Genetic and epigenetic changes have an important effect in modulating cancer proliferation and progression as well as immune resistance. We analyzed the association between ACOT7 expression and mutations and CNA via cBioPortal [34,35]. As shown in Figure 4A, the mutation status of ACOT7 in different tumors was evaluated. The deep deletion and amplification of the ACOT7 gene represent one of the critical factors leading to mutations, especially in ovarian epithelial tumors, esophagogastric adenocarcinoma, PAAD, SARC, and ESCA. In these cancers, the trend of ACOT7 gene alteration was consistent with its mRNA expression level. In addition, ACOT7 expression was positively correlated with CNA in 25 of 33 cancers except CHOL, DLBC, LAML, PCPG, THYM, THCA, LGG, and KIRP (Figure 4B).

DNA methylation can control gene expression without altering the genome sequence [42]. In the present study, ACOT7 expression negatively correlates with the DNA methylation level in 28 of 33 cancers, except OV, DLBC, LAML, GBM, and CHOL (Figure 4C). In addition, the DNA methylation level of the ACOT7 promoter was lower in UCEC, BLCA, BRCA, CESC, COAD, LIHC, LUAD, LUSC, PCPG, and READ compared to adjacent normal tissues (Figure 4D–M). Thus, the abnormal increase in ACOT7 mRNA expression might be associated with genetic alterations and reduced DNA methylation levels.

### 3.4. Relationship between ACOT7 Expression and TME

To investigate the potential functions and pathways of ACOT7, we conducted GSEA analysis. The data indicated that ACOT7 was notably related to cell cycle and immune regulation pathways, especially in BLCA, PAAD, CESC, COAD, BRCA, PRAD, SKCM, TGCT, and UVM (Figure 5).

To investigate the biological functions of ACOT7, we further performed GSVA analysis. Figure 6 shows the pathways that were positively or negatively correlated with ACOT7. It showed that ACOT7 was closely associated with CD4 T cells, CD8 T cells, and other pathways, suggesting that ACOT7 may serve a significant role in the TME.

There is accumulating evidence that TMEs are critically involved in tumor occurrence and development [43]. To identify the relationship between the TME and ACOT7, we analyzed RNA-seq data from multiple solid tumors in the TCGA database. It suggested that in most cancer types, except UCC, ACOT7 mRNA expression is significantly associated with TME pathways, especially with DNA damage, repair, and TME score A pathways (Figure 7A). We then analyzed the relationship between ACOT7 and the infiltration levels of 25 immune cells (Figure 7B), with the highest relevant levels in BLCA (*n* = 18), BRCA (*n* = 17), HNSC (*n* = 15), KIRC (*n* = 20), LUAD (*n* = 15), STAD (*n* = 15), THCA (*n* = 24), and THYM (*n* = 17) (Appendix A).

We further conducted an analysis of ACOT7 expression and immune-related genes, as described in Charoentong’s study [44]. Major histocompatibility complex (MHC), immune activation, immune-suppressive, chemokine, and chemokine receptor genes were collected and analyzed based on the TISIDB database. The heatmaps of immune-related gene expression patterns revealed that these genes were closely associated with ACOT7 expression in most tumor types (*p* < 0.05) (Figure 8). Collectively, ACOT7 may have a critical function in co-regulating the TME.

TMB and MSI are considered as indicators of immunotherapy response and prognosis. In the present study, ACOT7 expression correlated with TMB and MSI in multiple cancers, although all observed correlations are actually negligible according to Mukaka’s study [45]. ACOT7 expression was positively correlated with TMB in COAD (r = 0.14, *p* = 0.021), LIHC (r = 0.11, *p* = 0.045), PRAD (r = 0.13, *p* = 0.005), SKCM (r = 0.11, *p* = 0.018), STAD (r = 0.14, *p* = 0.004), and UCEC (r = 0.27, *p* < 0.001), while negatively correlated with TMB in LAML (r = −0.19, *p* = 0.037) (Figure 9A). In addition, ACOT7 expression was positively correlated with MSI in CESC (r = 0.12, *p* = 0.042), COAD (r = 0.17, *p* = 0.005), GBM (r = 0.19, *p* = 0.023), LIHC (r = 0.11, *p* = 0.038), LUSC (r = 0.09, *p* = 0.044), STAD (r = 0.17, *p* < 0.001), UCEC (r = 0.21, *p* = 0.005), and UVM (r = 0.23, *p* = 0.05) (Figure 9B). It is well known that immune-checkpoint-related genes are of great significance in immune escape [19]; we thus hypothesized that the expression of ACOT7 correlates with these genes. The findings support our speculation. ACOT7 expression was closely associated with most immune-checkpoint-associated genes, especially in BLCA, COAD, KICH, LIHC, and THCA (Figure 9C).

### 3.5. Drug Screening of ACOT7 in the GDSC Database

In addition to checkpoint blockade therapy, we tried to explore the correlation between ACOT7 and the sensitivity, risk, and efficacy of commonly used targeted therapeutic agents through the GDSC database. With the increased expression of ACOT7, the IC50 concentrations of targeted therapeutics, including afatinib (*p* < 0.001), gefitinib (*p* < 0.001), ibrutinib (*p* < 0.001), lapatinib (*p* = 0.0021), osimertinib (*p* < 0.001), lapatinib (*p* < 0.001), and taseletinib (*p* < 0.001), increased, while the IC50 concentration of PLX-4720 (*p* < 0.001) decreased (Figure 10). These results suggest that ACOT7 may be a potential predictive biomarker of targeted therapy sensitivity.

### 3.6. Knockdown of ACOT7 Inhibits Proliferation and Progression of Lung Cancer Cells

To further explore the role of ACOT7 in lung cancer, we performed cell function assays. The analysis of the CCLE database showed that ACOT7 expression was high in non-small-cell lung cancer (NSCLC) cells, including LUAD and LUSC (Figure 11A). According to the qRT-PCR assay, ACOT7 expression was higher in PC-9, A-549, and NCI-H1975 cell lines than in HBE cells, with the highest expression in PC-9 cells (Figure 11B). Therefore, the PC9 cell line were selected for further experimental validation. The expression level of ACOT7 was significantly suppressed when transfected with siRNA-ACOT7 (Figure 11C). In addition, the scratch assay indicated that the migratory capacity was reduced (Figure 11D), and the CCK8 assay showed the proliferation rate of PC9 cells was dramatically reduced (Figure 11E). In addition, Transwell and Matrigel experiments confirmed that the downregulation of ACOT7 significantly reduced the migration and invasion of PC9 cells (Figure 11F).

## 4. Discussion

Recently, immune checkpoint therapy has become quite popular, given its efficacy in many cancers [46]. However, a tiny percentage of patients are sensitive to immune checkpoint inhibitor therapy, and drug resistance and relapse are common [47]. Hence, further exploration is warranted to identify predictive biomarkers for immune checkpoint inhibitors. Pan-cancer analysis can indicate the regulation characteristics of molecular targets and provide insights into cancer diagnosis and therapy [48].

An increasing number of pan-cancer analyses have revealed that cancer driver genes, mutations, and RNA alterations are associated with tumorigenesis and development [49,50,51]. ACOT7 is a neuron-enzyme-regulating fatty acid metabolism, which plays a role in mesial lobe epilepsy [52]. However, the role of ACOT7 in pan-cancer and whether ACOT7 could be a prognostic indicator in lung cancer are still unknown. To date, studies investigating the role of ACOT7 in cancers are seldom. Xie et al. have reported that ACOT7 is activated by KLF13 and promotes the progression of hepatocellular carcinoma [53]. Research by Feng et al. has shown that gastric cancer patients with high ACOT7 expression have a lower survival rate [54]. In our study, ACOT7 was aberrantly expressed in various cancers through TCGA and HPA databases. High ACOT7 expression was significantly correlated with poorer prognoses of various cancers. GSEA analysis showed that ACOT7 was notably related to the cell cycle and TME pathway. In addition, functional experiments revealed that the downregulation of ACOT7 dramatically inhibited the proliferation, invasion, and metastasis ability of PC9 cells.

Increasing evidence supported TMB as a new and promising prognostic marker for cancers [55,56]. These studies demonstrated that a high TMB was related to an immunogenic TME and increased expression of neo-antigens, which can activate immune cells and further enhance the clinical response to immunotherapy. Interaction between tumor cells and the TME offers a new direction in anti-cancer immunotherapy. TME characteristics can be used as an indicator for the evaluation of the response of tumor cells to immunotherapy and are undergoing preclinical and clinical development [57]. In this study, ACOT7 mRNA expression was significantly related to DNA damage and repair pathways. Moreover, according to ESTIMATE scores, ACOT7 was involved in immune cell infiltration in most cancers. A TME with an infiltrating immune and extracellular matrix undergoes substantial changes that can impact tumor occurrence and development [58,59]. Our findings also suggested that ACOT7 was significantly associated with genes encoding tumor immunity factors. Previous studies reported that the viability and function of cancer cells and immune cells depend on the cellular metabolism, which is heavily influenced by the TME [60,61,62]. ACOT7 might alter the metabolism of cancer cells by regulating TME-related pathways. These results altogether clarify that ACOT7 has more vast applicability in pan-cancer and confirms that ACOT7 is a new therapeutic target for developing immunosuppressants.

An opposite trend was observed in the correlation between the expression of ACOT7 and the expression of many immune activation/suppression/MHC genes in some tumors. The activation of the tumor immune microenvironment can simultaneously inhibit its occurrence.. Some studies have confirmed that the level of infiltration of cancer-promoting immune cells and cancer-suppressing immune cells is significantly and positively correlated in tumor tissues [63]. The detailed mechanism needs to be further investigated.

Drug sensitivity has always been the focus of clinical discussion, and acquired drug resistance often leads to treatment failure and disease progression. To date, more than three generations of epidermal growth factor receptor–tyrosine kinase inhibitors (EGFR-TKIs) have been invented and applied in clinical practice. A meta-analysis of nine randomized clinical trials indicated that adjuvant EGFR-TKIs could improve the DFI of patients with EGFR-mutant NSCLC [64]. Despite their effectiveness, the management of acquired resistance to EGFR-TKIs remains a significant challenge [65]. In our study, ACOT7 expression was positively correlated with cellular sensitivity to EGFR-TKIs, and therefore, ACOT7 is potentially a powerful predictive indicator.

We successfully evaluated the role of ACOT7 from the pan-cancer perspective and further investigated the biological function of ACOT7 in LUAD. This is the first study regarding this topic, but on the other hand, the function of ACOT7 in more cancers needs to be further verified.

## 5. Conclusions

Taken together, our study first unveiled a complicated role of ACOT7-aberrant expression in clinical prognosis, immune cell infiltration, TMB, or MSI in pan-cancer. ACOT7 might act as an oncogene and potential therapeutic target in LUAD. As this is a preliminary study, further in vitro and in vivo experiments on the function of ACOT7 are needed.

## Figures and Tables

**Figure 1 cancers-14-04522-f001:**
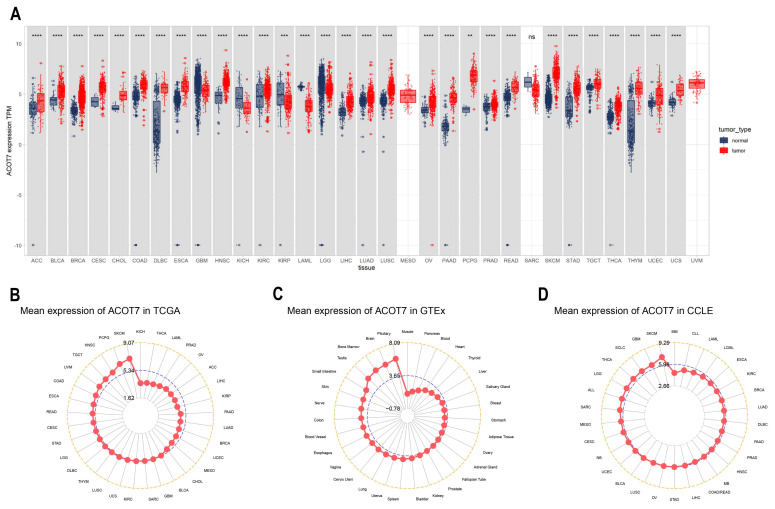
Differential expression of ACOT7 in human cancers. (**A**) Comparison of ACOT7 expression between tumors and normal samples from TCGA and GTEx datasets. (**B**) ACOT7 expression in 33 cancers from TCGA dataset. (**C**) ACOT7 expression in normal tissues from GTEx dataset. (**D**) ACOT7 expression in tumor cells from CCLE dataset. ** *p* < 0.01, *** *p* < 0.001, **** *p* < 0.0001, ns: not significant.

**Figure 2 cancers-14-04522-f002:**
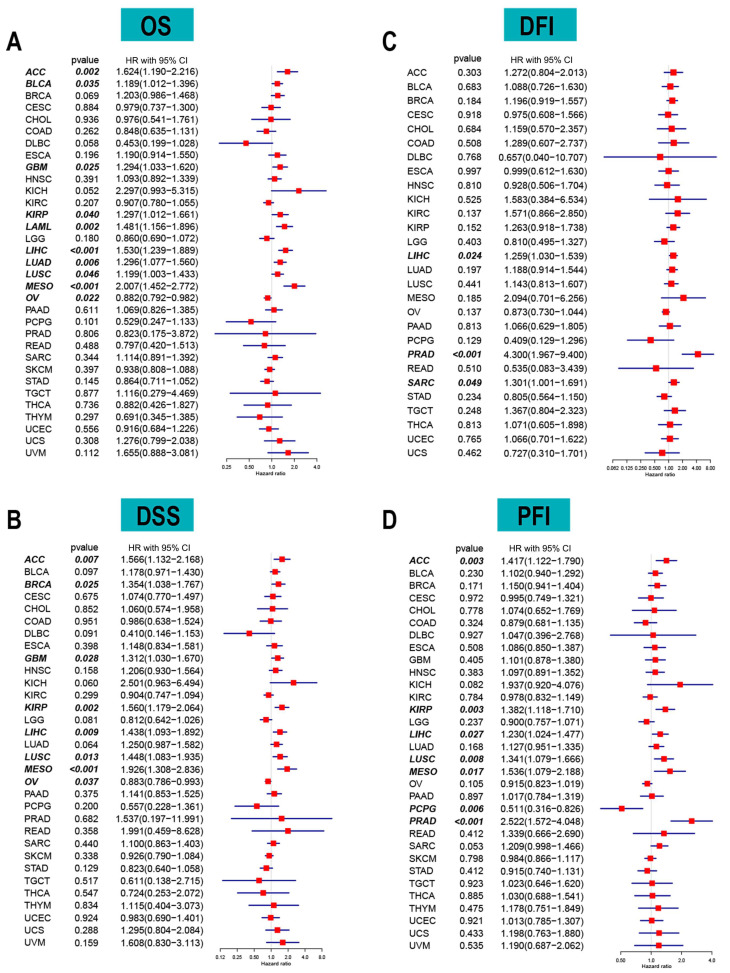
Forest plots based on univariate Cox regression analyses to show the relationship between ACOT7 mRNA expression and (**A**) overall survival (OS), (**B**) disease-free interval (DFI), (**C**) disease-specific survival (DSS), and (**D**) progression-free interval (PFI).

**Figure 3 cancers-14-04522-f003:**
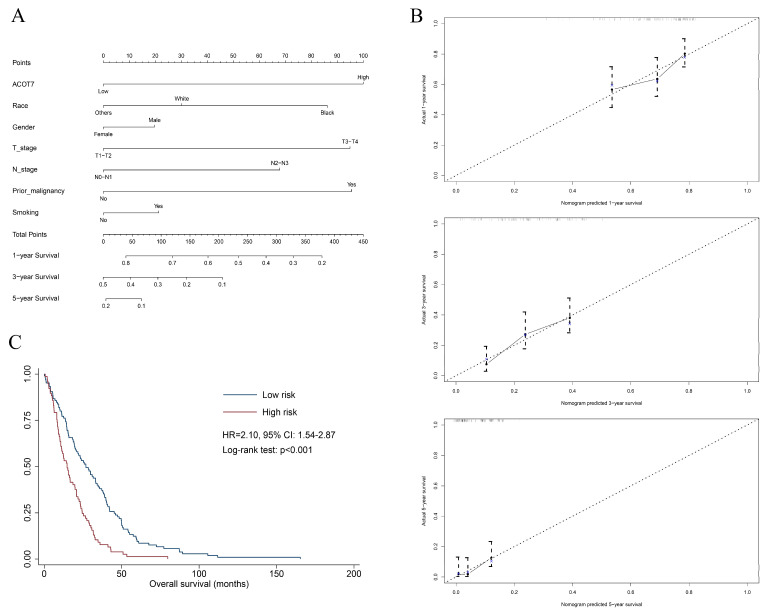
Nomogram based on multivariate Cox regression model to predict overall survival of LUAD patients using TCGA dataset. (**A**) Nomogram; (**B**) calibration plots of 1-, 3-, and 5-year overall survival; (**C**) Kaplan–Meier curve of patients in different risk groups (high-risk: score > 225; low-risk: score ≤ 225). Abbreviations: ACOT7, Acyl-CoA thioesterase 7; LUAD, lung adenocarcinoma.

**Figure 4 cancers-14-04522-f004:**
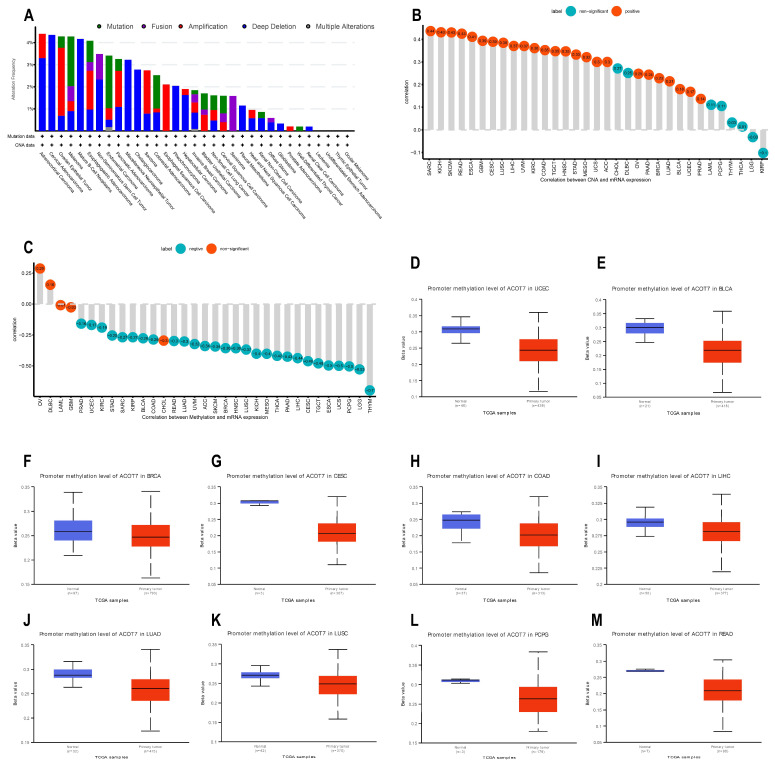
Genetic alteration analyses of ACOT7. (**A**) CNA and mutation frequency data of ACOT7 from cBioPortol dataset; (**B**) Pearson’s correlation between ACOT7 expression and CNA from TCGA dataset; (**C**) Pearson’s correlation between ACOT7 mRNA expression and DNA methylation from TCGA dataset; (**D**–**M**) comparison of promoter DNA methylation status of ACOT7 between cancer and adjacent normal tissues in (**D**) UCEC, (**E**)BLCA, (**F**) BRCA, (**G**) CESC, (**H**) COAD, (**I**) LIHC, (**J**) LUAD, (**K**) LUSC, (**L**) PCPG, and (**M**) READ. Abbreviations: ACOT7, Acyl-CoA thioesterase 7; LUAD, lung adenocarcinoma; CNA, copy number alteration.

**Figure 5 cancers-14-04522-f005:**
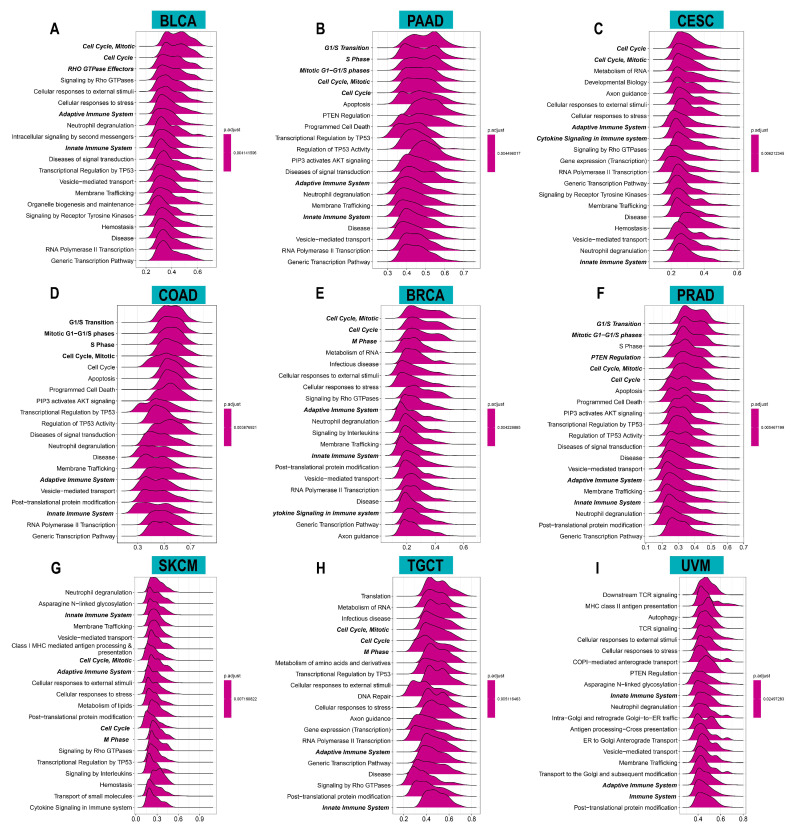
Results of gene set enrichment analysis (GSEA) in (**A**) BLCA, (**B**) PAAD, (**C**) CESC, (**D**) COAD, (**E**) BRCA, (**F**) PRAD, (**G**) SKCM, (**H**) TGCT, and (**I**) UVM. The *X*-axis presents the enrichment score.

**Figure 6 cancers-14-04522-f006:**
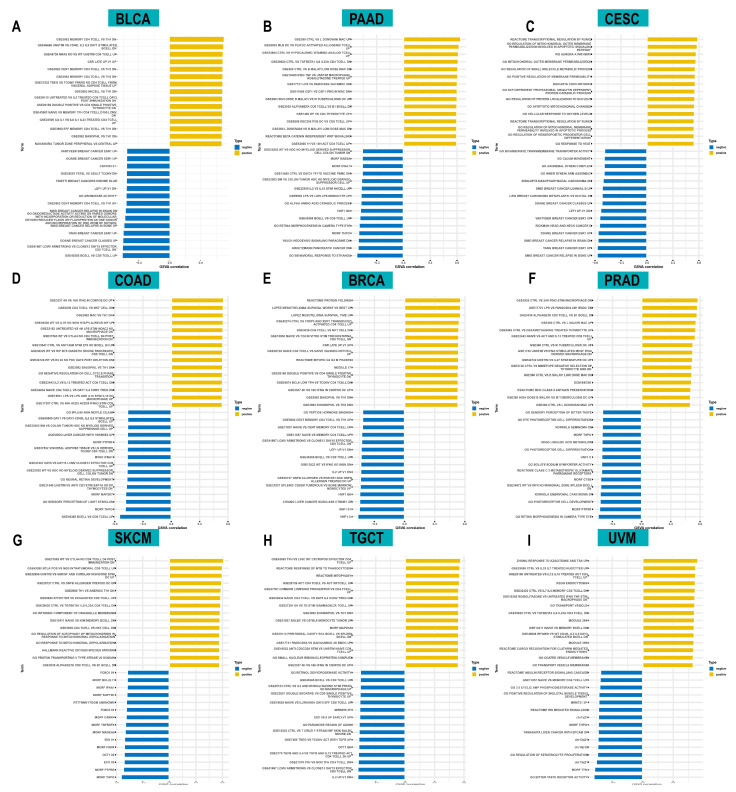
Results of gene set variation analysis (GSVA) in (**A**) BLCA, (**B**) PAAD, (**C**) CESC, (**D**) COAD, (**E**) BRCA, (**F**) PRAD, (**G**) SKCM, (**H**) TGCT, and (**I**) UVM.

**Figure 7 cancers-14-04522-f007:**
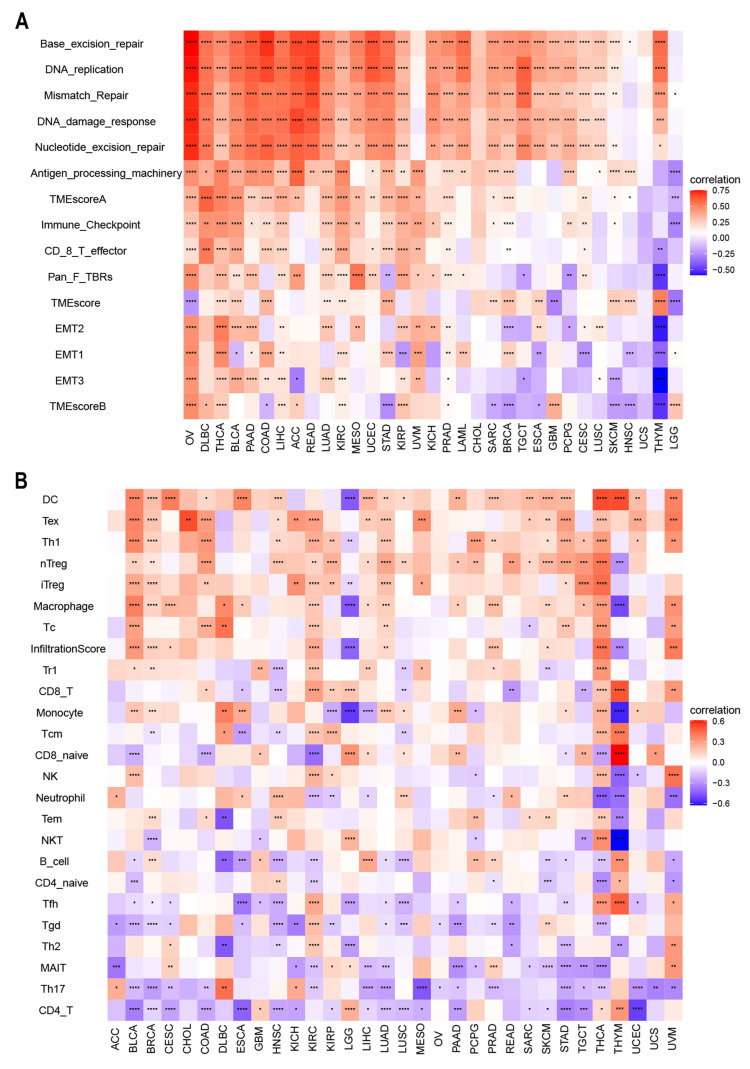
(**A**) Heatmaps to show the relationship between ACOT7 expression and tumor microenvironment (TME); (**B**) heatmaps to show the relationship between ACOT7 expression and immune cell infiltration. * *p* < 0.05, ** *p* < 0.01, *** *p* < 0.001, **** *p* < 0.0001.

**Figure 8 cancers-14-04522-f008:**
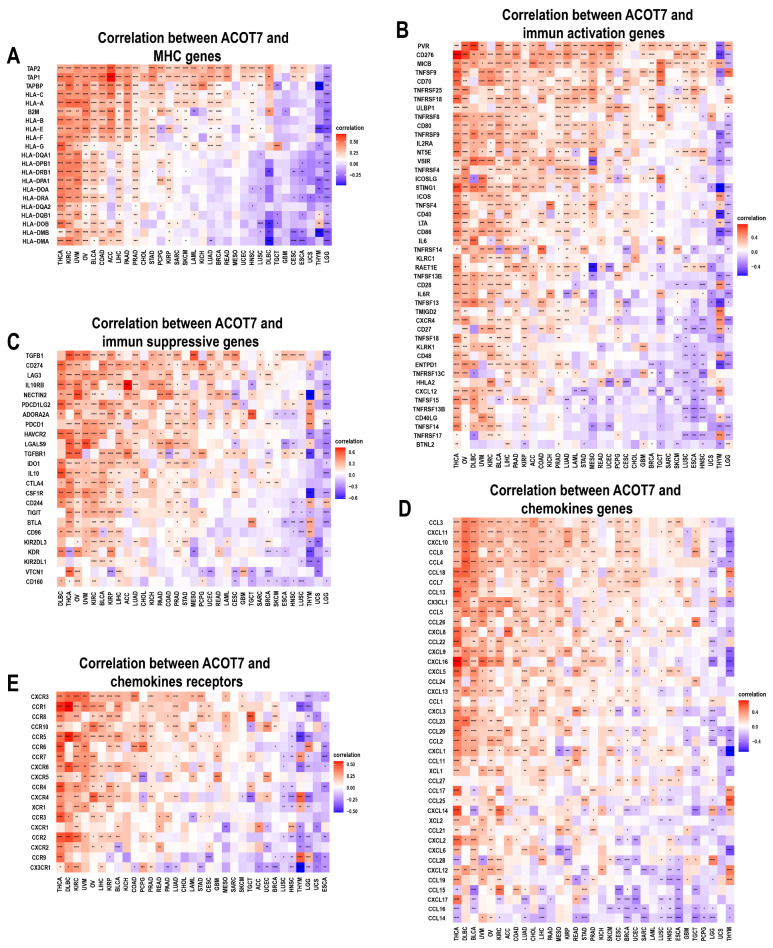
Heatmaps of ACOT7 and immune-related gene co-expression. (**A**) Major histocompatibility complex (MHC) genes, (**B**) immune activation genes, (**C**) immune suppressive genes, (**D**) chemokine genes, (**E**) chemokine receptor genes. * *p* < 0.05, ** *p* < 0.01, *** *p* < 0.001 **** *p* < 0.0001.

**Figure 9 cancers-14-04522-f009:**
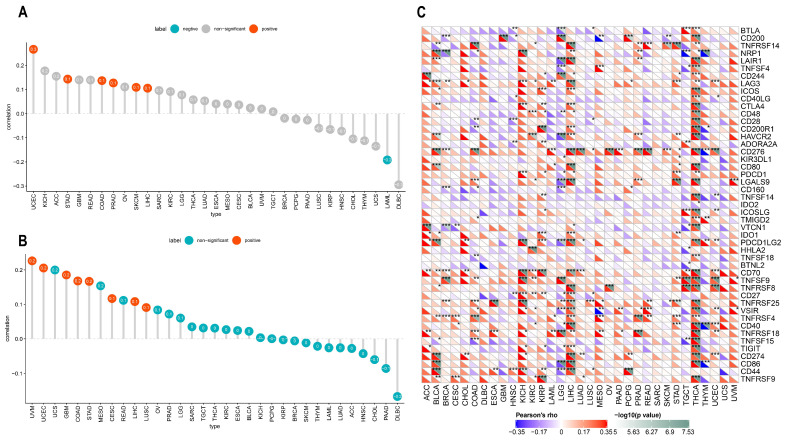
Pearson’s correlation between ACOT7 expression and (**A**) TMB and (**B**) MSI. (**C**) The heatmaps to show the relationship between ACOT7 expression and immune-checkpoint-associated genes. * *p* < 0.05, ** *p* < 0.01, *** *p* < 0.001.

**Figure 10 cancers-14-04522-f010:**
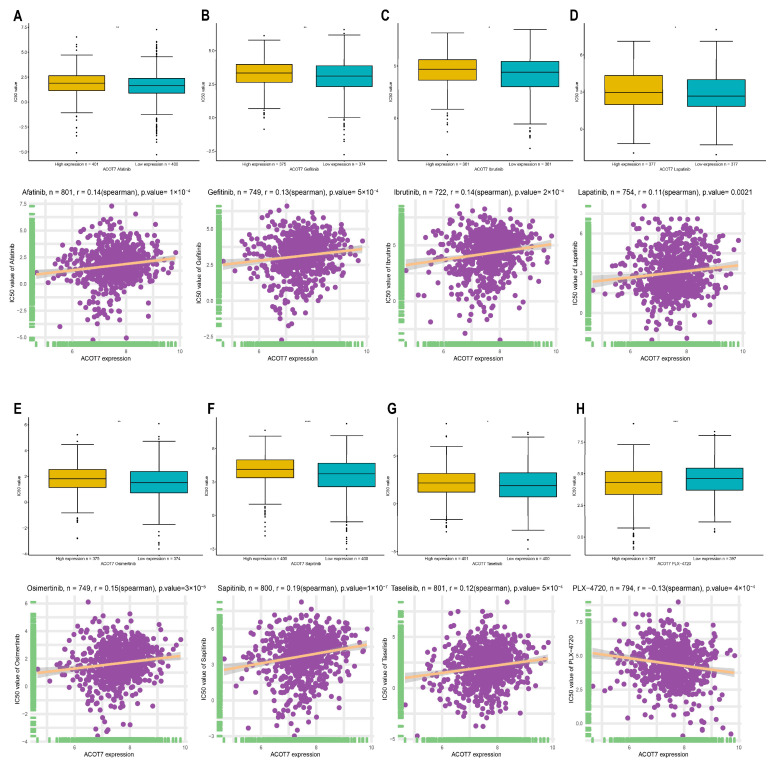
Spearman’s correlation between ACOT7 expression and drug sensitivity of (**A**) afatinib, (**B**) gefitinib, (**C**) ibrutinib, (**D**) lapatinib, (**E**) osimertinib, (**F**) sapitinib, (**G**) taselisib, and (**H**) PLX-4720. The comparison of IC50 concentrations between differential expression of ACOT7 were conducted using *t*-test. * *p* < 0.05, ** *p* < 0.01, *** *p* < 0.001, **** *p* < 0.0001.

**Figure 11 cancers-14-04522-f011:**
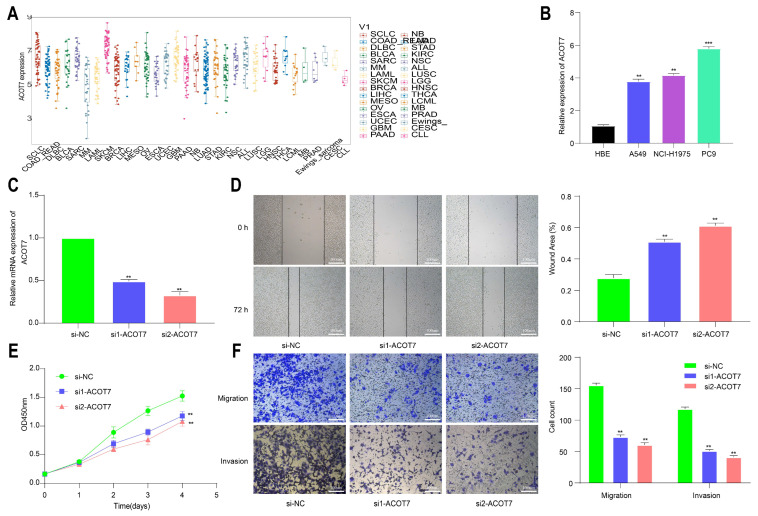
Knockdown of ACOT7 inhibited proliferation and progression in lung cancer cells. (**A**) ACOT7 expression in cancer cells via CCLE. (**B**) The expression level of ACOT7 in HBE and various lung cancer cell lines via qRT-PCR. (**C**) Transfection of PC9 cells with si-ACOT7 and the efficacy was verified via qRT-PCR. (**D**) Scratch assays. (**E**) CCK-8 assays. (**F**) Transwell and Matrigel assays. ** *p* < 0.01, *** *p* < 0.001.

**Table 1 cancers-14-04522-t001:** Univariate and multivariate Cox regression analyses of ACOT7 and clinical characteristics in LUAD from TCGA database.

Variables	Univariate		Multivariate	
	HR with 95% CI	*p* Value	HR (95%)	*p* Value
ACOT7 (high vs. low)	1.59 (1.07–2.36)	0.021	1.72 (1.17–2.60)	0.007
Race				
White	Reference		Reference	
Black	1.22 (0.75–2.00)	0.422	1.39 (0.82–2.30)	0.197
Others	0.87 (0.54–1.39)	0.552	0.79 (0.48–1.30)	0.363
Age (≥65 vs. <65, y)	0.89 (0.67–1.21)	0.475		
Gender (male vs. female)	1.17 (0.87–1.57)	0.291	1.11 (0.81–1.52)	0.526
T stage (T3–T4 vs. T1–T2)	1.85 (1.26–2.72)	0.002	1.71 (1.12–2.13)	0.013
N stage (N2–N3 vs. N0–N1)	1.57 (1.09–2.42)	0.014	1.46 (0.99–2.13)	0.052
M stage (M1 vs. M0)	1.10 (0.64–1.88)	0.732		
Prior malignancy	1.56 (1.04–2.33)	0.031	1.75 (1.15–2.66)	0.009
Smoking	1.05 (0.77–1.43)	0.772	1.16 (0.83–1.62)	0.390

Abbreviations: ACOT7, Acyl-CoA thioesterase 7; LUAD, lung adenocarcinoma; TCGA, the Cancer Genome Atlas; HR, hazard ratio; CI, confidence interval.

## Data Availability

The original contributions presented in the study are included in the article/Appendix A.

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
