# Peer review of "Pan-Cancer Analysis and Experimental Validation Identify ACOT7 as a Novel Oncogene and Potential Therapeutic Target in Lung Adenocarcinoma"

_cancers, 2022, doi:10.3390/cancers14184522_

Round 1

Reviewer 1 Report

Zheng and colleagues conducted a comprehensive analysis on ACOT7 expression levels across several tumor types and correlate it with immune and clinical features. They report that ACOT7 is commonly over-expressed in many tumors compared to respective normal tissues and high expression in tumors is also associated with lower survival. The authors also find correlations with immune cell infiltration and expression of various immune-related genes in some tumors. Lastly, they show that knocking down ACOT7 in PC9 lung cancer cells reduces cell proliferation, migration, and invasion.

The manuscript includes many different analyses and yields possibly interesting and abundant data. Additionally, the authors attempt to extend their computational findings with cell culture assays, which is appreciated. There are, however, some weaknesses and limitations (outlined below) that should be addressed to strengthen the manuscript.

Major comments

1. The presentation of the results should be more detailed in the text, materials and methods, and/or figure legends. It is not always obvious what was done or what the authors want to compare or found. This is most notable in Figure 5, where it is not clear what is on the X-axis here (adjusted p value I think), is it showing the histogram of the adj. p value for a given pathway/function in all TCGA/CCLE cases with gene expression data? I would also appreciate it if the authors could provide more details about how they performed their survival analysis. Were high and low ACOT7 expression groups used for KM curves and, if so, how were such groups defined? 

2. It is possible that rather than driving the TME, immune, and clinical features described in this paper that ACOT7 is upregulated as a consequence of other genomic alterations that occur in tumors. The authors show that ACOT7 is only mutated, amplified, or deleted in a small percentage of cancers yet ACOT7 expression is associated with increasing CNAs. So, an increase in ACOT7 expression could be the result of tumor aneuploidy. It seems plausible that aneuploid cells may favor upregulation of ACOT7 since their metabolic requirements are different from euploid, non-tumor cells. Similarly, ACOT7 expression is correlated with TMB, and cells with a high TMB may also favor upregulation of ACOT7 due to altered metabolic demands. This makes it difficult to say that ACOT7 is a prognostic marker when both TMB and aneuploidy are also known to be associated with patient survival. The authors should address the possibility of such confounding data. One possible way is to only compare within tumors that have similar CNA levels or TMB and determine whether survival is still lower in tumors with higher ACOT7 expression. The authors findings may indicate that highly mutated or aneuploid tumors could be more dependent on ACOT7 for survival/oncogenic functions, which would be very interesting and definitely warrant further research.

3. There are a few issues with the cell culture data. The authors should also show (or provide a reference) that protein expression correlates with mRNA expression in their cell lines. Similarly, they should show at the protein level that ACOT7 expression is depleted (minor issue). I also have an issue with their interpretation of their cell culture results. I think the phrase "validate the role of ACOT7 in lung cancer" is too strong. The observed decrease in proliferation, migration, and invasion after ACOT7 depletion could be an indirect effect of how the gene knockdown affects other processes (e.g., fatty acid and/or glucose metabolism). It would be interesting, and more direct, if the authors could show that upregulating ACOT7 in non-transformed lung cells alters cell functions and is able to increase tumor-like behaviors such as migration/invasion. Can the authors also please provide a reference that the scratch wound assay is predictive of metastatic potential (some normal/non-transformed cell lines may be highly migratory) or please revise metastatic to migratory in the text (line 284).

Minor comments

1. It is interesting that the correlation between ACOT7 expression and the expression of many immune activation/suppressive/MHC genes shows the opposite trends in some tumors. Can the authors please speculate in the discussion why this may be the case?

2. I think the authors should include more information about the role of metabolic alterations in cancer and the link to altered metabolism and the TME/tumor niche construction either in the introduction or discussion or both. PMID: 314348913099016833921421, 28202506, 2081423927386546, 32632251

3. If possible, any sort of assay (chemokine/cytokine ELISA panel) that could further establish the link between ACOT7 expression and the computational findings of immune correlates/features would be highly compelling.

Author Response

Dear reviewer,

Thanks a lot for your review! We have carefully read all your comments and revised our manuscript! We hope it can meet with your approval!

Kind regards,

Yi Shen

Reviewer 2 Report

In the submitted manuscript Zheng et al. presented results of their pan-cancer analysis which identify ACOT7 as a novel prognostic and immunological biomarker for lung adenocarcinoma.

This manuscript and study behind it is mostly based on re-analyses of publicly available data, using well-known and commonly used databases and tools. In that light this manuscript is non-inventive, mostly descriptive, since this type of manuscripts are being massively published nowadays. The only real added value are in vitro studies, which were presented almost en passant. However, this manuscript suits to the "Cancer Informatics and Big Data" section of this journal.

Nevertheless, this manuscript has many both major and minor drawbacks which must be corrected or explained and further improved.

1) The title "Pan-cancer analysis and experimental validation identify ACOT7 as a novel prognostic and immunological biomarker in lung adenocarcinoma" is misleading since authors with their in vitro experiments did not actually validated ACOT7 role as "prognostic and immunological biomarker in lung adenocarcinoma". They eventually showed that it is oncogene and potential therapeutic target for LUAD.

2) This study is to a great deal irreproducible because methods were described too poorly or not at all:

- for ALL used databases provide their valid web-addresses and cite references

- for ALL used tools and R-packages provide their version numbers and cite references if published in scientific journals

- describe in more details how differential expression of ACOT7 was analyzed, since just stating "using the “limma” package of R software" is not enough

- it is unclear how Figures 1B-D were created

- provide Resource Identification Initiative IDs from Cellosaurus database for all used cell lines, and always use proper cell line name (e.g., NCI-H1975, not just 1975)

- omit mentioning and results obtained on NCI-H1299 ("1299") cell line since this is a model for lung large cell carcinoma, not LUAD

- provide name of kit for reverse transcription, GAPDH primers sequences, qPCR cycling conditions, and cite reference for 2^ddCt method: https://doi.org/10.1006/meth.2001.1262

- state actual models and manufacturers of ALL used instruments: microplate reader, microscope, qPCR machine, etc.

- state which statistical test was used for testing normality of distribution ("t-test or Wilcoxon test was used as appropriate"), and state which values were presented on graphs in Figure 11 (means + SD or something else)

- for all HRs mentioned in the text provide their 95% CIs

- explain ALL methods whose results were presented in 'Results' section, since for instance usage of TISIDB and GDSC databases, and scratch assay were not explained in 'Methods'

3) There are lots of unclear and imprecise used words and sentences which must be explained in the text:

Lines 74-75: Sentence "Finding a novel target to compensate for the shortage of PD-1/PD-L1 antibodies is a promising approach to improving advanced cancer recurrence." is incomprehensible because it is unclear what means "shortage of PD-1/PD-L1 antibodies" and why would anybody want to "improve advanced cancer recurrence"?!

Line 155: it is unclear what means "highly" expressed

Line 157: it is unclear what means "in all 31 tumor cell lines" since there exist much more than 31 tumor cell lines

Line 171: it is unclear what means "dramatically" since HRs ranged from 0.882 to 2.007

Line 184: it is unclear what means "a fair predictive accuracy"

Lines 196-198: it is unclear what means "Mutation analysis showed that amplification of the ACOT7 gene is one of the critical factors leading to mutations"

Whenever in text you state that ACOT7 is correlated with something, like in lines 200-2001 "ACOT7 was positively correlated with CNA", state precisely with which particular ACOT7 feature (I suppose in that sentence you meant ACOT7 expression)

4) Figures are of very low quality:

- almost all figures have too small text and thus are unreadable

- in figure captions and table names all abbreviations presented in figures and tables must be explained

5) Table 2: it is unclear why non-significant variants from univariate analysis were used in multivariate analysis

6) Lines 250-256: precisely state which correlation coefficient was presented (r or rho) and clearly state that ALL observed correlations are actually negligible correlations (see Table 2 in https://www.ncbi.nlm.nih.gov/pmc/articles/PMC3576830/)

Minor errors:

Line 22: "comprehensive" instead of "compressive"

Line 54: Jade K is not the first author of Reference 1

Line 56: correct is "target for therapies"

Line 72: "still" is redundant

Line 143: 2 should be in subscript

Line 262: "commonly used" was repeated

In Figure 2 explain what are those numbers in parentheses presented after hazard ratio

Name of Supplementary Table 2 is incomplete.

Author Response

(The authors gave the same response as above.)

Round 2

Reviewer 1 Report

I applaud the authors for their thorough efforts in responding to the comments. I feel the manuscript has improved considerably and is acceptable for publication.

Author Response

Dear reviewer,

We all authors appreciate a lot for your great efforts. It is your professional and valuable comments that make our manuscript improved. Thanks for your patience again!

Yi Shen

Reviewer 2 Report

Authors have satisfactorily answered to my questions but haven't completely applied all my requests so there is still things which must corrected or extended to further improve quality of this manuscript.

1) For ALL used databases (TCGA, GTEx, CCLE...) cite their original reference. Only by this authors would have benefit from your usage of their efforts.

2) For ALL used R-packages (limma, survival, forestplot...) provide version numbers so your bioinformatic analyses could be fully reproducible.

3) It is unclear what sentence "Images were captured using a fluorescence microscopy using an inverted microscope (Nikon Eclipse 2000, Nikon)." is doing in "2.8. RNA extraction" section, and not in "2.9. Cell proliferation and migration/invasion assays".

4) At least original reference for TCGA-LUAD should be cited: https://www.nature.com/articles/nature13385

Author Response

Dear reviewer,

We all authors appreciate a lot for your great efforts. We have re-revised our manuscript according to your comments. Please kindly review the details in the files "Round 2_Response to reviewer #2.docx" and "Revised manuscript.docx". Thanks a lot for your patience again!

Yi Shen
